# HOW TO 0WN NAS IN YOUR SPARE TIME

**Sanghyun Hong, Michael Davinroy[†], Yiğitcan Kaya, Dana Dachman-Soled, Tudor Dumitraş**
University of Maryland, College Park
shhong@cs.umd.edu, michael.davinroy@gmail.com, yigitcan@cs.umd.edu,
danadach@ece.umd.edu, tdumitra@umiacs.umd.edu

## ABSTRACT

New data processing pipelines and novel network architectures increasingly drive the success of deep learning. In consequence, the industry considers top-performing architectures as intellectual property and devotes considerable computational resources to discovering such architectures through neural architecture search (NAS). This provides an incentive for adversaries to steal these novel architectures; when used in the cloud, to provide Machine Learning as a Service (MLaaS), the adversaries also have an opportunity to reconstruct the architectures by exploiting a range of hardware side channels. However, it is challenging to reconstruct novel architectures and pipelines without knowing the computational graph (e.g., the layers, branches or skip connections), the architectural parameters (e.g., the number of filters in a convolutional layer) or the specific pre-processing steps (e.g. embeddings). In this paper, we design an algorithm that reconstructs the key components of a novel deep learning system by exploiting a small amount of information leakage from a cache side-channel attack, Flush+Reload. We use Flush+Reload to infer the trace of computations and the timing for each computation. Our algorithm then generates candidate computational graphs from the trace and eliminates incompatible candidates through a parameter estimation process. We implement our algorithm in PyTorch and Tensorflow. We demonstrate experimentally that we can reconstruct MalConv, a novel data pre-processing pipeline for malware detection, and ProxylessNAS-CPU, a novel network architecture for the ImageNet classification optimized to run on CPUs, without knowing the architecture family. In both cases, we achieve 0% error. These results suggest hardware side channels are a practical attack vector against MLaaS, and more efforts should be devoted to understanding their impact on the security of deep learning systems.

## 1 INTRODUCTION

To continue outperforming state-of-the-art results, research in deep learning (DL) has shifted from manually engineering features to engineering DL systems, including novel data pre-processing pipelines (Raff et al., 2018; Wang et al., 2019) and novel neural architectures (Cai et al., 2019; Zoph et al., 2018). For example, a recent malware detection system MalConv, with a manually designed pipeline that combines embeddings and convolutions, achieves 6% better detection rate over previous state-of-the-art technique without pre-processing (Raff et al., 2018). In addition to designing data pre-processing pipelines, other research efforts focus on neural architecture search (NAS)—a method to automatically generate novel architectures that are faster, more accurate and more compact. For instance, the recent work of ProxylessNAS (Cai et al., 2019) can generate a novel architecture with 10% less error rate and 5x fewer parameters than previous state-of-the-art generic architecture. As a result, in the industry such novel DL systems are kept as trade secrets or intellectual property as they give their owners a competitive edge (Christian & Vanhoucke, 2017).

These novel DL systems are usually costly to obtain: generating the NASNet architectures (Zoph et al., 2018) takes almost 40K GPU hours and the MalConv authors had to test a large number of failed designs in the process of finding a successful architecture. As a result, an adversary who wishes to have the benefits of such DL systems without incurring the costs has an incentive to steal them. Compared to stealing a trained model (including all the weights), stealing the *architectural*

---

This work was done when Michael Davinroy was a research intern at the Maryland Cybersecurity Center.

*details* that make the victim DL system novel provides the benefit that the new architectures and pipelines are usually applicable to multiple tasks. Training new DL systems based on these stolen details still provides the benefits, even when the training data is different. After obtaining these details, an attacker can train a functioning model, even on a different data set, and still benefit from the stolen DL system (So et al., 2019; Wang et al., 2019). Further, against a novel system, stealing its architectural details increases the reliability of black-box poisoning and evasion attacks (Demontis et al., 2019). Moreover, stealing leads to threats such as Camouflage attacks (Xiao et al., 2019) that trigger misclassifications by exploiting the image scaling algorithms that are common in DNN pre-processing pipelines.

The emerging Machine-Learning-as-a-Service (MLaaS) model that offers DL computation tools in the cloud makes remote hardware side-channel attacks a practical vector for stealing DL systems (Liu et al., 2015). Unlike prior stealing attacks, these attacks do not require physical proximity to the hardare that runs the system (Batina et al., 2019; Hua et al., 2018) or direct query access to train an approximate model (Tramèr et al., 2016). Cache side-channel attacks have especially been shown as practical in cloud computing for stealing sensitive information, such as cryptographic keys (Liu et al., 2015). Cache side-channel attacks are ubiquitous and difficult to defeat as they are inherent to the micro architectural design of modern CPUs (Werner et al., 2019).

In this paper, considering the incentives to steal a novel DL system and applicability of cache side-channel attacks in modern DL settings, we design *a practical attack to steal novel DL systems by leveraging only the cache side-channel leakage*. Simulating a common cloud computing scenario, our attacker has a co-located VM on the same host machine as the victim DL system, and shares the last-level cache with the victim (Liu et al., 2015). As a result, even though the VMs are running on separate processor cores, the attacker can monitor the cache accesses a DL framework—PyTorch or TensorFlow—makes while the victim system is running (Liu et al., 2015).

The first step of our attack is launching a cache side-channel attack, Flush+Reload (Yarom & Falkner, 2014), to extract a single trace of victim's function calls (Section 3). This trace corresponds to the execution of specific network operations a DL framework performs, e.g., convolutions or batch-normalizations, while processing an input sample. However, the trace has little information about the computational graph, e.g., the layers, branches or skip connections, or the architectural parameters, e.g., the number of filters in a convolutional layer. The limited prior work on side-channel attacks against DL systems assumed knowledge of the architecture family of the victim DNN (Yan et al., 2018; Duddu et al., 2018); therefore, these attacks are only able to extract variants of generic architectures, such as VGG (Simonyan & Zisserman, 2015) or ResNet (He et al., 2016). To overcome this challenge, we also extract the approximate time each DL operation takes, in addition to the trace, and we leverage this information to estimate the architectural parameters. This enables us to develop a *reconstruction algorithm* that generates a set of candidate graphs given the trace and eliminates the incompatible candidates given the parameters (Section 4). We apply our technique to two exemplar DL systems: the MalConv data pre-processing pipeline and a novel neural architecture produced by ProxylessNAS.

**Contributions.** We design an algorithm that reconstructs novel DL systems only by extracting cache side-channel information, that leaks DL computations, using Flush+Reload attack. We show that Flush+Reload reliably extracts the trace of computations and exposes the time each computational step takes in a practical cloud scenario. Using the extracted information, our reconstruction algorithm estimates the computational graph and the architectural parameters.

We demonstrate that our attacker can reconstruct a novel network architecture found from NAS process (ProxylessNAS) and a novel manually designed data pre-processing pipeline (MalConv) with no reconstruction error.

We demonstrate the threat of practical stealing attacks against DL by exposing that the vulnerability is shared across common DL frameworks, PyTorch and TensorFlow.

## 2 BACKGROUND

Here, we discuss prior efforts in both crafting and stealing network architectures. There is a growing interest in crafting novel DL systems as they significantly outperform their generic counterparts. The

immense effort and computational costs of crafting them, however, motivates the adversaries to steal them.

**Effort to Design Deep Learning Systems.** Creating deep learning systems traditionally takes the form of human design through expert knowledge and experience. Some problems require novel designs to manipulate the input in a domain-specific way that DNNs can process more effectively. For example, MalConv malware detection system (Raff et al., 2018) uses a manually designed pre-processing pipeline that can digest raw executable files as a whole. Pseudo LIDAR (Wang et al., 2019), by pre-processing the output of a simple camera sensor into a LIDAR-like representation, achieves four times better object detection accuracy than previous state-of-the-art technique. More-over, recent work also focuses on automatically generating optimal architectures via neural architecture search (NAS). For example, reinforcement learning (Zoph & Le, 2016) or gradient-based approaches (Cai et al., 2019) have been proposed for learning to generate optimal architectures. Even though NAS procedures have been shown to produce more accurate, more compact and faster neural networks, the computational cost of the search can be an order of magnitude higher than training a generic architecture (Zoph et al., 2018).

**Effort to Steal Deep Learning Systems.** Prior work on stealing DNN systems focus on two main threat models based on whether the attacker has physical access to the victim's hardware. Physical access attacks have been proposed against hardware accelerators and they rely on precise timing measurements (Hua et al., 2018) or electromagnetic emanations (Batina et al., 2019). These attacks are not applicable in the cloud setting we consider. The remote attacks that are applicable in the cloud setting, on the other hand, have limitation of requiring precise measurements that are impractical in the cloud (Duddu et al., 2018). Further, the attack without this limitation (Hong et al., 2018) requires the attacker to know the family the target architecture comes from; thus, it cannot steal novel architectures. In our work, we design an attack to reconstruct novel DL systems by utilizing a practical cache side-channel attack in the cloud setting.

# 3 Extracting the Sequence of Computations via Flush+Reload

## 3.1 Threat Model

We consider an attacker who aims to steal the key components in a novel DL system, i.e., a novel pre-processing pipeline or a novel network architecture. We first launch a Flush+Reload (Yarom & Falkner, 2014) attack to extract cache side-channel information leaked by DL computation. Our target setting is a cloud environment, where the victim's DL system is deployed inside a VM—or a container—to serve the requests of external users. Flush+Reload, in this setting, is known to be a weak, and practical, side-channel attack (Liu et al., 2015). Further, as in MLaaS products in the cloud, the victim uses popular open-source DL frameworks, such as PyTorch (Benoit Steiner, 2019) or TensorFlow (Abadi et al., 2016).

**Capabilities.** We consider an attacker that owns a *co-located* VM—or a container—in the same physical host machine as the victim's system. Prior work has shown that spinning-up the co-located VM in the third-party cloud computing services does not require sophisticated techniques (Ristenpart et al., 2009; Zhang et al., 2011; Bates et al., 2012; Kohno et al., 2005; Varadarajan et al., 2015). Due to the co-location, the last-level cache (L3 cache) in the physical host is shared between multiple cores where the attacker's and victim's processes are; thus, our attacker can monitor the victim's computations leaked at the L3 cache. We also note that, even if the victim uses GPUs, our attacker can still observe the same computations used for CPUs via cache side-channels (see Appendix A).

**Knowledge.** We consider our attacker and the victim use the same version of the same open-source DL framework. This is realistic, in MLaaS scenarios such as AWS SageMaker or Google Cloud's AutoML, as cloud providers recommend practitioners to use the common frameworks to construct their systems. These common practices also allow our attacker to reverse-engineer the frameworks offline and identify the lines of code to monitor with the Flush+Reload technique.

---

For example, AWS provides convenient deployment options for both PyTorch and TensorFlow: `https://docs.aws.amazon.com/sagemaker/latest/dg/pytorch.html`, and `https://docs.aws.amazon.com/sagemaker/latest/dg/tf.html`.

| Online (Co-located) | Offline (Separate) |
|---|---|
| ② Monitor the lines of code via Flush+Reload | ① Identify the lines of code to monitor

③ De-noise the Flush+Reload observations
④ Profile the computations with a set of parameters
⑤ Perform the reconstruction process with the data |

Figure 1: **Overview of our attack procedure.** Our attacker only requires to be online (co-located) while the attacker is monitoring the computations of a victim DL system.

## 3.2 FLUSH+RELOAD MECHANISM

Flush+Reload allows an adversary to continually monitor victim's instruction access patterns by observing the time taken to load them from memory. This technique is effective to extract the computation flow of the victim's program when the attacker and victim share memory (i.e., a shared library or page deduplication (Bosman et al., 2016)). The attacker flushes specific lines of code in a shared DL framework from the co-located machine's cache-hierarchy and then measure the amount of time it takes to reload the lines of code. If the victim invokes the monitored line of code, the instruction will be reloaded into the shared cache, and when the attacker reloads the instruction, the access to it will be noticably faster. On the other hand, if the victim does not call the monitored line of code, the access to it will be slower because the instruction needs to be loaded from main memory (DRAM). By repeating this process, our attacker can tell when a victim has accessed a line of code.

## 3.3 OVERVIEW OF OUR ATTACK PROCEDURE

In Figure 1, we illustrate our attack procedure. We split the steps into two phases: the online phase and the offline phase. In the online phase (step ②), the attacker needs co-location to monitor the computations from the victim's system. In the offline phase (steps ①, ③, ④, and ⑤) attacker does not require the co-location with the victim.

① First, our attacker analyzes the open-source DL framework to identify the lines of code to monitor. The attacker monitors the first line of each function that corresponds to the start of a DL computation.
② Next, the attacker spins up a co-located VM and launches the Flush+Reload attack to extract the trace of victim system's function calls. As the trace does not depend on the input sample, we only require to extract a single trace from one full invocation of the victim system.
③ Since the raw observations with Flush+Reload are noisy, the attacker applies filtering to highlight the regularities of DL computations reflected in the trace.
④ To estimate the architectural parameters, e.g., the input/output channels, kernel size, or strides, our attacker creates a lookup tables of timings and performed number of matrix multiplications by collecting traces from various parameter combinations.
⑤ Finally, using the victim's computational trace lookup tables for estimating architectural parameters, the attacker starts the reconstruction process to steal the victim's DL system (Sec 4).

## 3.4 MONITORING THE TOY NETWORK COMPUTATIONS VIA FLUSH+RELOAD

**Experimental Setup.** We implement our attack on Ubuntu 18.04 running on a host machine equipped with the Intel E3-1245v6 3.7GHz processors (8 cores, 32GB memory and 8MB cache shared between cores). For the step ①, we analyze two popular open-source DL frameworks, PyTorch and TensorFlow, and identify the list of functions to monitor (see Appendix C for the full list of functions). We leverage the Mastik toolkit (Yarom, 2016) to launch the Flush+Reload attack, and while a victim DL system is running on a VM, our attacker monitors the list of functions—step ②. For the reconstruction process conducted in offline after the extraction, we use Python v3.6 to implement the procedure.

---

https://www.python.org

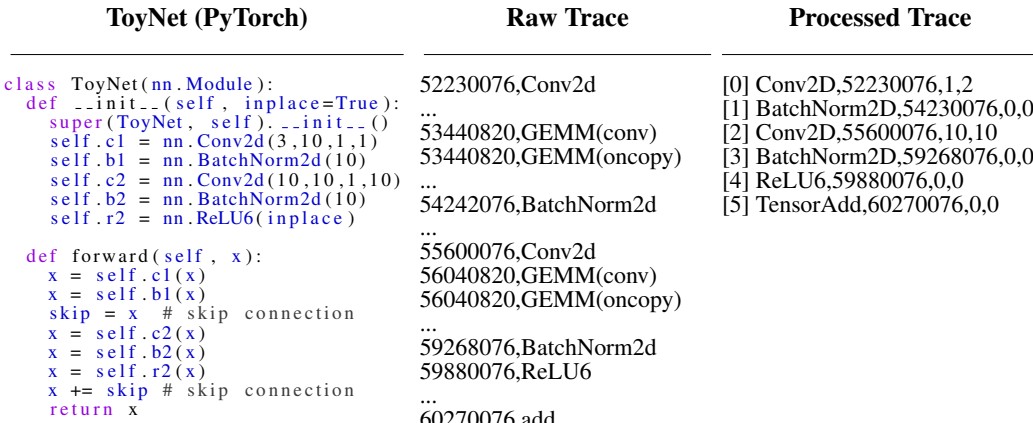

Figure 2: **Extracted traces while ToyNet is in use.** In the left, we place our network definition in PyTorch. We show the raw trace from Flush+Reload (middle) and the de-noised trace (right).

**ToyNet Results.** In Figure 2, we demonstrate the extracted trace via Flush+Reload while ToyNet is processing an input. ToyNet is composed of one convolution followed by a batch-norm and one depthwise convolution followed by a batch-norm and a ReLU activation. The 1st convolution has the parameters (in, out, kernel, stride) as $(3, 10, 3, 1)$, and the depthwise convolution's parameters are $(10, 10, 1, 1)$. The network has a skip connection that adds the intermediate output (from the 1st convolution) to the final output. During inference, we feed in an input with dimensions 3x32x32.

In the middle panel of Figure 2, we also show the raw—noisy—trace from the Flush+Reload output. The trace only includes cache-hits where the attacker's accesses to the lines of code are faster, i.e., when the victim invokes the function. Each element of the trace includes a timestamp and a function name. The name corresponds to the ToyNet layers, such as Conv2d and BatchNorm2d, and it also contains additional information such as the tensor (add) and the BLAS operations, e.g., GEMM(oncopy).

Our attacker filters the raw trace according to the regular patterns in the DL computation. For example, a long function call, e.g., Conv2d in the ToyNet trace, can appear multiple times in the trace, as the cache can hit multiple times during Flush+Reload. In this case, we condense the multiple occurrences into a single invocation using a heuristic based on how close the timestamps are. We also observe the matrix multiplications such as GEMM(conv) and GEMM(oncopy) while DL computation is being processed. We count the individual occurrences and sum them up them based on the timestamp. After obtaining the processed trace (in the right panel), the attacker starts the reconstruction procedure.

## 4 RECONSTRUCTING NOVEL DEEP LEARNING SYSTEMS

After processing the Flush+Reload trace, our attacker reconstructs the key components of the victim's DL system. In this process, the attacker aims to *generate the candidate computational graphs* of the victim system and to *eliminate the incompatible candidates* by estimating the correct parameter set for each computation. For instance, in our ToyNet example, the attacker wants to identify the computational orders and the location of the start and end of a branch connection (computational graph). Also, the same attacker wants to estimate the parameters for each computation; for example, the input/output channels and the kernel size in the 1st Conv2d. In this small network that has one branch, there are only 10 candidate computational graphs; however, considering all possible combinations of parameters, this will result in untractable number of candidates. Prior work, in reconstruction, only considered generic architectures such as VGGs or ResNets with the unrealistic assumption that an attacker knows the architecture family (*backbone*); however, as our aim is to steal novel DL systems, we do not make this assumption. To overcome this problem, we design a reconstruction procedure, which we describe next.

**Knowledge of Our Attacker in Reconstruction.** Here, we consider our attacker knows what tensor operations and functions to monitor in the victim's open-source DL framework. These functions are model-independent; they correspond to architectural attributes designated by the deep learning framework (see Appendix C). We show that this knowledge is sufficient to reconstruct novel data-preprocessing pipelines, such as MalConv, that are usually shallower than the network architectures.

To reconstruct the deeper network architecture automatically designed by NAS algorithms, we assume our attacker has some knowledge about the NAS search space—e.g., NASNet search space (Zoph et al., 2018)—the victim's search process relies on. This knowledge includes the list of layers used and the fact that a set of layers (known as blocks) are repeatedly used such as Normal and Reduction Blocks in NASNet. We make this assumption because, from the sequence of computations observed via Flush+Reload, our attacker can easily identify a set of layers and the repetitions of the layers. However, we do mot assume how each block is composed by using the layer observations directly; instead, we identify candidate blocks by using a sequence mining algorithm. We demonstrate that, under these assumptions, our attack reconstructs the ProxylessNAS-CPU in 12 CPU hours rather than running a NAS algorithm from scratch that takes 40k GPU hours.

## 4.1 OVERVIEW OF OUR RECONSTRUCTION PROCEDURE.

We first focus on the invariant rules in the computations used for DL computations. For instance, there are unary operations and binary operations. The tensor addition used to implement a skip connection is binary operation; thus, our attacker can supplement the reconstruction process by pruning the incompatible candidates. We also exploit the fact that computation time is proportional to the number of element-wise multiplications in a computation. In the ToyNet example, the time the 1st convolution (2 million cycles) takes is shorter than the 2nd depthwise convolution (3.668 million cycles); thus, our attacker further eliminates the candidates by comparing the possible parameters for a computation with her offline profiling data—the lookup table.

Our reconstruction procedures consist of two steps:

① *Generation*: The attacker can generate the candidate computational graphs from the Flush+Reload trace based on the invariant rules in DL computations. Using the rules, our attacker reduces the number of candidates significantly.

② *Elimination*: Our attacker compares the time for each computation takes with profiling data and prunes the incompatible candidates. We estimate the parameters sequentially starting from the input. When the output dimension from a candidate does not match with the observation, we eliminate.

**Error Metrics.** To quantify the error of our reconstruction result, we use two similarity metrics. First, we use the graph edit distance (GED) (Abu-Aisheh et al., 2015) to compare the reconstructed computational graph with that of the victim. Second, we use the $\ell_1$-distance to compute the error between the estimated architectural parameters and those in the victim system.

**Victims.** We first reconstruct the MalConv (Raff et al., 2018), a novel data pre-processing pipeline that converts the binary file into a specific format so that a neural network can digest easily. Also, we show that our attacker can reconstruct the novel ProxylessNAS (Cai et al., 2019) architecture that shows the improved accuracy on the ImageNet classification with the less computational cost on a CPU.

## 4.2 RECONSTRUCTING NOVEL PRE-PROCESSING PIPELINES

Here, we elaborate the reconstruction process of the MalConv (Raff et al., 2018) data pre-processing pipeline. MalConv receives the raw bytes of `.exe` files and determines whether the file is malicious or not. The uniqueness of MalConv comes from the way that it treats the sequence of bytes: 1) Code instructions in binary file are correlated spatially, but the correlation has discontinuities from function calls and jump commands that are difficult to capture by the sequence models, e.g., RNNs.

---

Note that ProxylessNAS starts its searching process from a backbone architecture such as NASNet; thus, even if the paper reported a search took 200 GPU hours, this number does not include the time spent searching a backbone architecture, i.e., the 40k GPU hours to find NASNet.

| | | | | Reconstruction Errors | |
|---|---|---|---|---|---|
| Victim | Type | # Candidates | # Compatibles | Topology (GED) | Parameters ($\ell_1$) |
| MalConv | Pre-processing Pipeline | 20 | 1 | 0 | 0 |
| ProxylessNAS-CPU | Deep Neural Network | 180,224 | 1 | 0 | 0 |

Table 1: **Reconstruction results.** We report the number of candidates and compatible architectures. Also, we report the reconstruction errors in the resulting topology and parameters, and they are 0%.

2) Also, each sequence has the order of two million steps which far exceeds the length of an input to any previous neural network classifier. MalConv tackles this problem by pre-processing the sequence of bytes (Figure 3). It first splits the upper four bits and the lower four bits (narrow operations) of a byte information; this helps the network capture the locality of closer bytes and distant bytes. Next, the pipeline uses one dimensional convolution to extract such localities and performs the element-wise multiplications of two outputs. Before feeding this information to the neural network, the pipeline uses max-pooling to reduce the training time caused by processing inputs with large dimensions. All these heuristics are examined manually (see Section 4 of the original paper); thus, our attacker can save time and effort by stealing the pipeline.

**Generate Computational Graphs.** The first step of our attacker is to reconstruct the computational graph candidates for the victim pipeline from the Flush+Reload trace. As we can see in the trace in Figure 3, the attacker cannot simply connect the components in the traces sequentially because the branch connection, e.g., `[7] * (multiply)`. Also, from this component, our attacker knows which is the end of a branch but cannot know when the branch has started. We solve this problem by populating all possible candidates and pruning them later with the parameter estimation.

Our algorithm populates the candidate computational graphs and the sample candidates found (see Appendix E). Our solution uses a recursive algorithm. Given a trace from Flush+Reload ($T$), we pop each computation $t$ from the back and construct the list of candidates $l$. At a high-level, the algorithm first traverses all the possible connections starting from the last computation to the first by using recursion. Then, when the base condition is met (i.e., the algorithm arrives the first computation, Embeddings), we backtrack the recursions to construct the list of candidate computational graphs. We focus on the computation type in this backtracking process; there are unary and binary computations. For the unary operations, we simply connect the current and preceding computations. However, for the binary operations, we split all the preceding computations into a set of two lists. Each set of two lists corresponds to a branch, and we continue backtracking for each branch and include all of the construction into our results. At the end, *we found 20 candidates*.

**Eliminate Candidates with Computational Parameters.** Next, our attacker further prunes the candidates based on the computational parameter estimation process. Our attacker, most importantly, focuses on the fact that computation time is dependent on the size of the matrix multiplication. This enables our attacker to profile the computational time taken for a set of param-

**MalConv Novel Pre-processing Pipeline**    **Processed Trace**

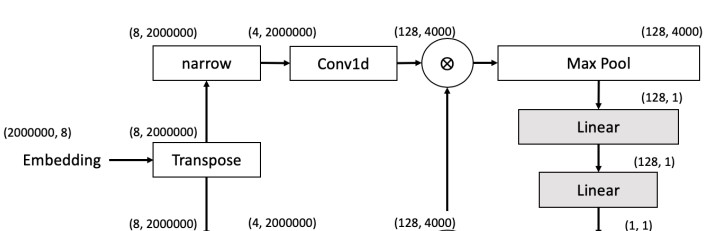

[0] Embedding,4,0
[1] transpose,210610,0
[2] narrow,210754,0
[3] Conv1d,941689,8371993
[4] narrow,9473620,0
[5] Conv1d,10144145,8355807
[6] Sigmoid,18655690.5,0
[7] * (multiply),18775738,0
[8] MaxPool1d,18835039,0
[9] transpose,18949264,0
[10] Linear,18949546,96
[11] Linear,18949831,0
[12] Sigmoid,18949905,0

Figure 3: **MalConv and the extracted trace.** While the MalConv (left) is processing a sample, we extract a trace via Flush+Reload and process the trace (right) for the pipeline reconstruction.

| Block Size | Counts | Identified Block |
|:---:|:---:|:---|
| 1 | 1 | AvgPool2d, Linear |
| 2 | 38 | Conv2d - BatchNorm2d |
| 3 | 19 | Conv2d - BatchNorm2d - ReLU |
| 4 | 5 | Conv2d - BatchNorm2d - Conv2d - BatchNorm2d |
| 5 | 17 | DepthConv2d - BatchNorm2d - ReLU6 - Conv2d - BatchNorm2d |
| 6 | 10 | DepthConv2d - BatchNorm2d - ReLU6 - Conv2d - BatchNorm2d - add |
| 7 | 0 | - |
| 8 | 15 | Conv2d - BatchNorm2d - ReLU6 - DepthConv2d - BatchNorm2d - ReLU6 - Conv2d - BatchNorm2d |
| 9 | 8 | Conv2d - BatchNorm2d - ReLU6 - DepthConv2d - BatchNorm2d - ReLU6 - Conv2d - BatchNorm2d - add |

Table 2: **The 9 candidate blocks identified from the Flush+Reload trace.**

eter combinations in advance. The attacker is able to perform this offline by taking advantage of cloud infrastructure: that the hardware and software stacks composing the cloud are consistent. In the MalConv reconstruction, we profile the timing of the convolution and linear operations. For the convolutions, we consider input/output channels $\{1, 2, 4, 8, 16, 32, 128, 256\}$, kernels $\{1, 3, 5, 7, 11, 100, 200, 500, 1k, 10k\}$, and strides $\{1, 2, 5, 10, 100, 200, 500, 1k, 10k\}$. For the linear layers, we use input $\{4, 8, 16, 32, 64, 128, 256, 512, 1024, 2048\}$ and output dimensions $\{1, 10, 16, 20, 32, 40, 100, 128, 256, 512, 1k, 1024, 2048\}$.

Once our attacker has the timing profiles with these parameter combinations, the attacker defines the potential parameter sets for the convolutions and linear layers. Then, the attacker checks, in each candidate, if the computational graph returns the correct output dimension (1,) for the input (8, 2000000). In this pruning process, there are the other operations such as Sigmoid, * (multipy), transpose, narrow, or pooling. We applied the universal rules for each case: 1) the Sigmoid and multiply do not change the input/output dimensions, 2) the transpose only swaps two dimensions in an input, 3) the narrow slice one chosen dimension, e.g., (8,2000000) to (4,1000000); thus we consider all the possible slice in checking, and 4) the pooling only requires us to estimate its window size, so we match this value to the stride of a preceding convolution. At the end of this parameter estimation, *we can narrow down to only one architecture with the correct set of computational parameters, i.e., 0% error.*

## 4.3 RECONSTRUCTING NOVEL NETWORK ARCHITECTURES

Here, we show our attacker is able to steal a novel network architecture by describing the reconstruction process of the ProxylessNAS-CPU (Cai et al., 2019) that improves the accuracy of existing architecture, MobileNetV2 (Sandler et al., 2018), and also reduces the computation time. Indeed, the NAS search procedure warm-starts from an over-parameterized MobileNetV2 as a backbone; however, in our attack, *we hypothesize our attacker is not aware of the backbone*. Instead, we assume our attacker only knows the search space of MNasNet (Tan et al., 2019) (see Appendix D) where the authors come up with the MobileNetV2, opposed to the recent attacks in Sec 2.

Knowing the search space does not, however, reduce the amount of efforts by our attacker in reconstruction. The network architectures found from the NAS procedure commonly are wide and deep, and they include multiple branch connections; thus, our attacker requires to consider exponential number of candidate computational graphs and the computation parameters, which makes the attack infeasible. To tackle this issue, we focus on the NAS procedure—this process factorizes the entire architecture into blocks by their functions. For instance, NASNet (Zoph et al., 2018) is composed of normal cells (blocks) and reduction cells. Within each block, the process considers the architecture combinations that provides the optimal performance. Thus, we first identify the potential blocks before we initiate the process for reconstructing candidate computational graphs.

**Identifying Candidate Blocks.** We utilize the frequent subsequence mining (FSM) method to identify the blocks composing the ProxylessNAS-CPU architecture. Our FSM method is simple: we iterate over the Flush+Reload trace with the fixed windows and count the occurrences of each subsequence. Since the attacker knows that in the search space that the victim uses, a maximum of nine computations are used to compose a block, we consider the window size from one to nine. Once we count the number of occurrences for each subsequence (candidate blocks), and we prune

https://aws.amazon.com/ec2/instance-types/

them based on the rules in the search space: 1) a Conv2d operation is followed by a BatchNorm, 2) a block with a DepthConv2d must end with a Conv2d and BatchNorm (for a depthwise separable convolution), 3) a branch connection cannot merge (add) in the middle of the block, and 4) we take the most frequent block in each window. In Table 2, we describe the *9 identified blocks*. We then run the generation process of reconstructing candidate computational graphs with the blocks instead of using each computation in the trace. At the end, *we have 180,224 candidate computational graphs*.

| **Processed Trace** | **Identified Block** | **Estimated Parameters** |
|---|---|---|
| ... | ... | ... |
| [8] Conv2d,305693984,1,20 | [B-8] Conv2d,1,20 | $C_1$:24 in, 144 out channels |
| [9] BatchNorm2d,323455984,0,0 | [B-8] BatchNorm2d,0,0 | 144 in/out channels |
| [10] ReLU6,376113984,1,1 | [B-8] ReLU6,0,0 | |
| [11] DepthConv2d,426259984,143,205 | [B-8] DepthConv2d,143,205 | 144 in/out channels |
| [12] BatchNorm2d,585059984,0,0 | [B-8] BatchNorm2d,0,0 | 144 in/out channels |
| [13] ReLU6,592321984,0,0 | [B-8] ReLU6,0,0 | |
| [14] Conv2d,609547984,1,7 | [B-8] Conv2d,1,7 | 144 in, $C_2$:32 out channels |
| [15] BatchNorm2d,614331984,0,0 | [B-8] BatchNorm2d,0,0 | $C_2$:32 in/out channels |
| ... | ... | ... |

Figure 4: **The reconstruction of the ProxylessNAS-CPU architecture.** From the Flush+Reload trace (left), we find the candidate block (middle) and estimate the computation parameters (right).

**Eliminate Candidate with Computational Parameters.** For each candidate composed of known blocks, our attacker estimates the computation parameters. However, the number of parameter combinations are also exponential; for example, within the search space, a Conv2d can have any number of input/output channels, kernel size $\{1, 3, 5\}$, and strides $\{1, 2\}$. Thus, we focus on the computation rules in a block. 1) We first found that DepthConv2d is only possible to have the same input/output channels. Also, the channel size can be identified by the number of GEMM(conv) operations. For instance, in Figure 4, the DepthConv2d has 143 GEMM(conv) invocations, which is close to the channel size. Since commonly the operation has an even number of channels, the attacker can easily reduce the candidates to 142 or 144. 2) We also know that the number of GEMM(oncopy) invocations is proportional to the matrix multiplication size in a Conv2d; thus, the attacker can compare the offline profiling results with the processed traces and estimate the parameters. For instance, the 1st Conv2d has 20 GEMM(oncopy), and we approximately have a set of input dimensions, e.g., (20-30,112,112) from the previous block estimation. Thus, our attacker only profiles the variations of input channels $\{20 - 30\}$, kernels $\{1, 3, 5\}$, and strides $\{1, 2\}$—total 60 cases and check if there is a match. Moreover, 3) the Conv2d after DepthConv2d is the pointwise linear operation whose kernel and stride is one, which will further reduce the attacker's efforts. Our attacker runs this elimination process and *finally narrows down to only one architecture with the correct set of computational parameters, i.e., 0% error*.

## 5 DISCUSSION

In this section, we discuss defense mechanisms that prevent our attacker from reconstructing the victim's DL system with an exact match. Prior work on defenses against cache side-channel attacks proposed system-level solutions (Kim et al., 2012; Liu et al., 2016; Werner et al., 2019). However, applying them requires infrastructure-wide changes from cloud providers. Also, even if the infrastructure is resilient to cache side-channel attacks, an attacker can leverage other attack vectors to leak similar information. Thus, we focus on the defenses that can be implemented in DL frameworks.

We design our defense mechanisms to obfuscate what the attacker observes via cache side-channels by increasing the noise in computations supported by DL frameworks. We discuss four approaches that blend noise into components of a DL framework; however, these mechanisms introduce a computational overhead by performing additional operations. This highlights that defending against our attack is not trivial and efficient countermeasures require further research.

**Padding Zeros to the Matrix Multiplication Operands.** Our reconstruction algorithm estimates the computational parameters such as kernel sizes or strides based on the time taken for matrix multiplication. Hence, we consider increasing the size of operands randomly by padding zeros to

them. We keep the original sizes of the operands and, after the multiplication of augmented tensors, we convert the resulting tensor into that of the correct dimensions by removing the extra elements. With the augmentation, our attacker finds it difficult to reconstruct the victim's DL system exactly by monitoring a single query. However, if our attacker can observe computations with multiple queries, the attacker can cancel-out the noise and estimate the parameters correctly.

**Adding Null/Useless Network Operations.** This reconstruction attack assumes all the computations observed in the Flush+Reload trace are used to compute the output of a DL system. Thus, a defender can modify the victim's architecture so that it includes the identity layers or the branches whose outputs are not used. We hypothesize a small number of null/useless operations will not increase the attacker's computational burden significantly; this addition only increases the time needed to reconstruct the victim's architecture by a few hours. If the defender includes an excessive amount of null/useless layers or branches, this can significantly increase the reconstruction time. However, this defense suffers from two issues: 1) the defense may still not make the reconstruction impossible, and 2) the victim also requires to perform the additional operations, which increases network evaluation time significantly.

**Shuffling the Computation Order.** We have seen in popular DL frameworks that, once a network architecture is defined, the computational order of performing operations is also invariant. We are able to shuffle the computation order of the victim's DL system each time when the system processes an input. In particular, we can identify the dependency of operations in a victim's DL system and compute the independent operations in a different order each time. This approach will make the observations from cache side-channels inconsistent, which results in the exponential number of candidate architectures that our attacker needs to consider. However, to compute the independent operations separately, the defender needs to store intermediate results in memory while processing an input; thus, this approach increases the space overhead of the DL computations.

**Running Decoy Operations in Parallel.** Lastly, we can make a DL framework run separate networks (decoy operations) in parallel on the same physical host. These networks obfuscate what our attacker will observe via Flush+Reload. Here, the attacker cannot reconstruct the victim architecture by monitoring a single query because the computational order does not reflect how the victim's architecture is defined. However, if our attacker can observe the computations over multiple queries, the attacker can use the frequent sequence mining (FSM) that we used in the block identification to identify a repeated set of operations and can reconstruct the victim architecture. This defense also increases network evaluation time by running extra operations on the same machine.

## 6 Conclusions and Future Work

This work presents an attack that reconstructs a victim's novel DL system through the information leakage from a cache side-channel, Flush+Reload. We steal key components of the victim's system: a novel pre-processing pipeline and a novel network architecture. Observing the DL computations and the time to complete each computation enables the attacker to populate all candidate computational graphs and prune them with our parameter estimation process. In experiments, we demonstrate the feasibility of this reconstruction attack by reconstructing MalConv, a novel pre-processing pipeline for malicious file detection, and ProxylessNAS-CPU, a novel architecture for the ImageNet classification optimized to run on CPUs. We do this with 0% error. As novel DL systems become trade secrets, our results highlight the demands for future work on countermeasures against model theft.

### Acknowledgments

We thank the anonymous reviewers for their valuable feedback. This research was partially supported by the Department of Defense, by NSF grants #CNS-1933033, #CNS-1840893, #CNS-1453045 (CAREER), by a research partnership award from Cisco and by financial assistance award 70NANB15H328 from the U.S. Department of Commerce, National Institute of Standards and Technology. We would like to thank the NSF REU-CAAR program (NSF grant #CCF-1560193).

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

## A    APPLICABILITY TO GPUS

Our attack is not fundamentally different for GPUs. In most deep learning frameworks, when a network performs a computation, it invokes the same function implemented in C++ and the function decides whether the back-end computation can use GPUs or not. This practice maximizes the hardware compatibility of a framework; however, this also makes the framework vulnerable to our attacker who can still observe the common functions listed in Table 3 by monitoring the shared cache. On GPUs the timings would be different, so we would have to profile the computational times, e.g., the time taken for the matrix multiplication with various sizes of tensor operands. However, on both CPUs and GPUs, the computation time is proportional to the size of tensor operands, which enables our attacker to estimate the architecture parameters with timing observations.

## B    DL COMPUTATIONS MONITORED IN PYTORCH AND TENSORFLOW

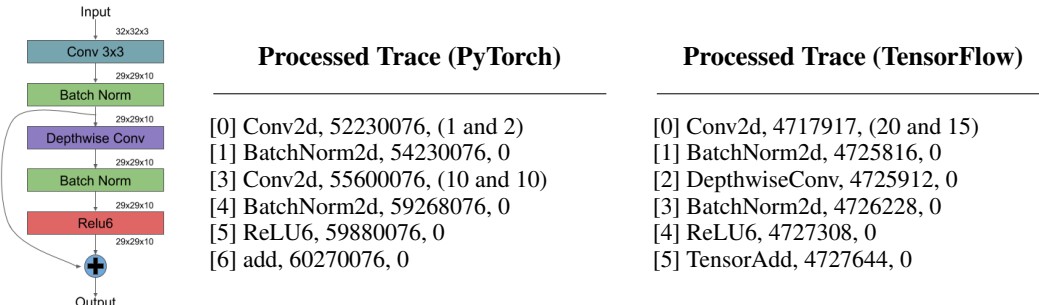

**Figure 5: The traces extracted from PyTorch and TensorFlow via Flush+Reload.** We illustrate the architecture of a small network (ToyNet) on the left. The sequence of computations observed from the our attack are listed in the middle (PyTorch) and the right (TensorFlow).

Figure 5 describes the reconstruction process of a small network in both the PyTorch and TensorFlow frameworks. On the left, we have the ground truth of the ToyNet architecture, which represents an example of a possible residual block in a victim network. In the middle and right, we show the observations of an adversary monitoring both PyTorch and TensorFlow code. The first entry indicates the monitored function corresponding to the desired architectural attribute. The second entry indicates the timestamp at which the adversary observes these functions, and the last entry is the number of general matrix multiplication (GEMM) function calls for the given layer observation.

Naming conventions vary slightly between the two frameworks, but the information inferred is the same. The adversary attacking both networks sees functions calls that correspond to architectural attributes in the same order: Conv2d, BatchNorm2d, Conv2d/DepthwiseConv, BatchNorm2d, ReLU6, and TensorAdd. PyTorch does not distinguish between Conv2d and DepthwiseConv, but as stated in 4.1, we can differentiate the layers by timing data. Additionally, PyTorch and TensorFlow use different linear algebra libraries to perform matrix computation, so the implementations differ slightly. However, they both use variations on matrix multiplication algorithms that take into account system level optimizations, such as cache size (e.g. Goto's algorithm). In both cases, we observe operations in nested iterations of these implementations and are able to monitor instructions that correspond to the size of the matrices being multiplied, giving an adversary the ability to estimate the parameters of the convolution layers.

To perform the estimations of these layer parameters, the adversary can profile candidates offline on similar hardware. They can then create a data set of candidate parameters for given observation ranges. For instance, the number of observed GEMM calls in the PyTorch example for the depthwise convolution layer gives the attacker the information that there are 10 output channels, and therefore also ten output channels in the 1st convolution. Additionally, the observed GEMM calls for the 1st convolution layer give the candidate kernel sizes of 3 and 5. Likewise in TensorFlow, the observed instructions fits the candidate kernel sizes of 3 or 5, and 0-24 output channels. Therefore, these

---

Customization of operations in TensorFlow: `https://www.tensorflow.org/guide/create_op`

| Framework | | Computations | Line of Code |
|---|---|---|---|
| **PyTorch** | Core (C++) | Conv2d | aten/src/ATen/native/LegacyNNDefinitions.cpp:49 |
| | | Conv1d | aten/src/ATen/native/Convolution.cpp:448 |
| | | BatchNorm | aten/src/ATen/native/Normalization.cpp:428 |
| | | FC (Begin) | aten/src/TH/generic/THBlas.cpp:329 |
| | | FC (End) | aten/src/TH/generic/THBlas.cpp:499 |
| | | ReLU6 | aten/src/ATen/LegacyTHFunctionsCPU.cpp:21320 |
| | | Sigmoid | aten/src/ATen/native/UnaryOps.cpp:191 |
| | | AvgPool | aten/src/ATen/native/AdaptiveAveragePooling.cpp:324 |
| | | MaxPool | aten/src/ATen/native/Pooling.cpp:50 |
| | | Embeddings | aten/src/ATen/native/Embedding.cpp:15 |
| | | add | aten/src/ATen/native/BinaryOps.cpp:46 |
| | | * (multiply) | aten/src/ATen/native/BinaryOps.cpp:88 |
| | | transpose | aten/src/ATen/natinsve/TensorShape.cpp:643 |
| | | narrow | aten/src/ATen/native/TeorShape.cpp:364 |
| | OpenBLAS | GEMM(conv) | driver/level3/level3.c |
| | | GEMM(oncopy) | kernel/x86_64/sgemm_ncopy_4_skylakex.c:54 |
| **TensorFlow** | Core (C++) | Conv2d | core/kernels/conv_2d.h:81 |
| | | BatchNorm | core/kernels/fused_batch_norm_op.cc:254 |
| | | DepthConv2d | core/kernels/depthwise_conv_op.cc:161 |
| | | FC | core/kernels/matmul_op.cc:542 |
| | | ReLU6 | core/kernels/relu_op_functor.h:68 |
| | | AvgPool | core/kernels/avgpooling_op.cc:80 |
| | | TensorAdd | core/kernels/cwise_ops_common.h:93 |
| | Eigen | #FCNodes | src/Core/products/GeneralMatrixVector.h:155 |
| | MKL-DNN | GEMM | src/cpu/gemm/gemm_driver.cpp:242 |
| | | (loop)M-outer | src/cpu/gemm/gemm_driver.cpp:349 |
| | | (loop)K | src/cpu/gemm/gemm_driver.cpp:355 |
| | | (loop)N | src/cpu/gemm/gemm_driver.cpp:372 |
| | | (loop)M-inner | src/cpu/gemm/gemm_driver.cpp:387 |

Table 3: **Monitored lines of code in PyTorch and TensorFlow.**

exploitable vulnerabilities exist independent of the specific deep learning framework a victim is using.

## C  LIST OF FUNCTIONS MONITORED VIA FLUSH+RELOAD

Table 3 shows the exact lines of code we monitor in the PyTorch and TensorFlow frameworks. We use PyTorch v1.2.0 and Tensorflow v1.14.0. In both the frameworks, we are able to monitor a similar set of DL computations in the C++ native implementations. However, the back-end libraries supporting the matrix multiplications are different, i.e., PyTorch is compiled with OpenBLAS whereas TensorFlow uses Eigen and MKL-DNN. Even if the libraries are different, the multiplications are implemented using GOTO's algorithm (Goto & Geijn, 2008). Therefore, we monitor the number of iterations of for-loops to estimate the overall size of a matrix multiplication.

## D  MNASNET SEARCH SPACE

Tan et al. (2019) utilize a hierarchical search space over six parameters: *ConvOp*, *KernelSize*, *SER-atio*, *SkipOp*, *FilterSize*, and *#Layers*. They choose to partition a CNN into a known, finite set of blocks and then further divide these blocks into possibly repeated layers. The number of repeats per layer in a given block $i$ is a searchable parameter $N_i$, which is bounded at $\pm 1$ the number of layers in MobileNetV2 on which block $i$ is based. These layers are further divided into three possible

---

https://github.com/pytorch/pytorch/commit/8554416a199c4cec01c60c7015d8301d2bb39b64
https://github.com/tensorflow/tensorflow/commit/87989f69597d6b2d60de8f112e1e3cea23be7298

network layers (*ConvOp*): regular convolution, depthwise convolution, or mobile inverted bottleneck convolution. Additionally, the network layer parameters can vary. These parameters include the convolution kernel size (*KernelSize*), the squeeze-and-excitation ratio (*SERatio*), a possible skip op (*SkipOp*), and the output filter size (*FilterSize*). The squeeze-and-excitation ration (*SERatio*) of a given layer varies between 0 and 0.025; the convolution kernel size varies between 3 and 5; the skip op is either pooling, identity residual, or no skip; and the filter size varies between 0.75, 1.0, and 1.25 the filter size of the corresponding block in MobileNetV2. Overall, this gives an claimed typical search space size of $10^{13}$ possibilities with 5 blocks, 3 average layers per block, and 432 options for the sub search space of a each block. This size compares to the per-layer approach with the same parameters that has a search space size $10^{39}$.

# E    SEARCHING CANDIDATE COMPUTATIONAL GRAPHS

**Generated Candidates**

**Algorithm 1** Populate computational graphs

```
 1: procedure POPULATEGRAPHS(T)
 2:     t = T.pop()
 3:     if t is empty then
 4:         g = CreateAGraph(t)
 5:         l = CandidateList()
 6:         l = l ∪ g
 7:         return l
 8:     else
 9:         if t is unary operator then
10:             l = PopulateGraphs(T)
11:             for g in l do
12:                 CreateAnEdge(t, g)
13:             end for
14:             return l
15:         else if t is binary operator then
16:             for each preceding element e in T do
17:                 Tl, Tr = Split(T, t)
18:                 ll = PopulateGraphs(Tl)
19:                 lr = PopulateGraphs(Tr)
20:                 for each gl, gr of ll × lr do
21:                     g = Compose(el, er)
22:                     l = l ∪ g
23:                 end for
24:             end for
25:             return l
26:         end if
27:     end if
28: end procedure
```

Figure 6: **The algorithm for searching candidate computational graphs.** We describe our algorithm to populate the candidate graphs of MalConv (left) and the sample candidates (right).

