# OpenReview forum: "How to 0wn the NAS in Your Spare Time"
_ICLR.cc/2020/Conference — Accept (Poster)_

### Official Review · AnonReviewer3 · 2019-10-23
**Official Blind Review #3**

**Rating:** 6

**Review:**

This paper proposes a way to attack and reconstruct a victim's neural architecture that is co-located on the same host. They do it through cache side-channel leakage and use Flush+Reload to extract the trace of victim's function call, which tells specific network operations. To recover the computational graph, they use the approximate time each operation takes to prune out any incompatible candidate computation graph. They show that they can reconstruct exactly the MalConv and ProxylessNAS.

The paper looks very interesting but also alarming -- more research should be done to countermeasure this attack. I have the following questions:

1. To reconstruct the network, you need to generate potentially exponentially number of candidates and do some pruning based on the estimated parameters. This also looks very expensive. I am wondering compared to just doing NAS yourself, how much gain in terms of resources and time this attack can give?

2. What is the limitation of the proposed approach, i.e., does it work on any network structures, e.g., sequence networks, graph convolutional networks, etc.

3. In the experiments shown, you can reconstruct MalConv and ProxylessNAS with zero error, does the proposed approach alway find the exact match? Under what circumstances can you find the exact match?

**Experience Assessment:**

I do not know much about this area.

**Review Assessment: Checking Correctness Of Derivations And Theory:**

I assessed the sensibility of the derivations and theory.

**Review Assessment: Checking Correctness Of Experiments:**

I assessed the sensibility of the experiments.

**Review Assessment: Thoroughness In Paper Reading:**

I read the paper at least twice and used my best judgement in assessing the paper.

---

> ### Author Response · Authors · 2019-11-13
> **Clarifications Regarding The Attacker’s Gain and The Limitations**
>
> We thank the reviewer for their constructive feedback. Here, we provide answers to your questions and concerns, and we updated our paper for clarification.
>
> (1) The Gain of the Attacker in terms of the Time and Resources
>
> Our reconstruction algorithm saves a significant amount of time and resources over running NAS search algorithms because the attack does not involve training of candidate architectures—i.e., the attacker only requires a CPU to run the reconstruction process, whereas running NAS demands multiple GPUs, and does the reconstruction in a shorter time. In our experiments, we reconstructed MalConv in a few minutes and ProxylessNAS-CPU in 12 hours on a CPU. Compared to the victim’s search time for one architecture, e.g., Proxyless-G (mobile) that takes approximately 40k GPU hours*, our attacker can steal the same architecture by only investing 0.003% of this time. We acknowledge the reviewer pointed out our attacker’s strength, and we incorporated this information in Sec 4 in our revised paper.
>
> (*) ProxylessNAS starts its searching process from a backbone architecture such as NASNet; thus, even if the paper reported a search took 200 GPU hours, this number does not include the time spent searching a backbone architecture, i.e., the 40k GPU hours to find NASNet.
>
> (2)/(3) The Limitations of Our Attack.
>
> Our attacker can reconstruct the architectures when they are implemented by using the computations (i.e., tensor operations and layers) in an open-source deep learning framework. For instance, EvolvedTransformer [1] is implemented in TensorFlow and utilizes the conventional layers and tensor operations; thus, our attacker can monitor all the computations via Flush+Reload. Additionally, an example GCN in PyTorch* uses the sequence of tensor multiplications and a bias addition to implement a new GraphConvolution layer. Therefore, we believe our attacker will be able to observe the sequence of these computations (e.g., torch.nn, torch.spmm, and a ‘+’ operation) via Flush+Reload and reconstruct networks using this layer.
>
> (*) https://github.com/tkipf/pygcn/blob/master/pygcn/layers.py#L9 (line 31)
>
> Our reconstruction attack relies on several assumptions, and we agree that our algorithm may not always find an exact match (0% error) quickly if these assumptions are not met. The limitations are as follows:
>
> 1) We assume that the victim architecture is implemented using the computations supported by an open-source deep learning framework. This includes that any layers a victim may create are based on these supported computations. However, if a victim uses a closed-source framework (i.e. a framework developed by and/or only available to them), it disables our attacker from monitoring computations via Flush+Reload. We believe, due to the popularity of open-source deep learning frameworks and the difficulty in developing and maintaining them, this is a rare scenario.
>
> 2) We also assume that the time taken for a tensor multiplication will depend on the size of tensor operands, and the computations will be performed in sequential order. If the computation time of an operation is not proportional to the size of operands, or the operations are performed out-of-order, the reconstructed architecture and its parameters will not match the victim’s architecture. However, we believe no common deep learning framework breaks these properties, and implementing them would likely be computationally inefficient.
>
> 3) A network with an excessive number of branches can explode the number of architecture candidates and increase the time taken for the reconstruction significantly. However, this type of architecture does not make our reconstruction attack impossible, and having this property is not common due to the significant addition to the computational overhead introduced.
>
> [References]
> [1] The Evolved Transformer, ICML’19

---

### Official Review · AnonReviewer1 · 2019-10-23
**Official Blind Review #1**

**Rating:** 3

**Review:**

Summary
---
This paper proposes to use a computer security method, "Flush+Reload" to infer the DNN architecture of a victim in the setting where both the attacker and the victim share the same machine in a cloud computing scenario. This does not require any physical access to the machine, however it does require that a CPU is shared, and the inference of the architectural details is based on the time it takes to reload computations from cache.
The paper is overall clear and well written.

Motivations of the paper
---
However, concerning the motivations of the paper, I'd like some clarifications. As far as I know, in the deep learning community, the most effective architectures are published and public (VGG, Inception, ResNet, Transformer...).
I am a bit confused by the sentence "As a result, in the industry such novel DL systems are kept as trade secrets or
intellectual property as they give their owners a competitive edge (Christian & Vanhoucke, 2017)." which justifies that architectures are kept secret and thus may be prone being stolen.
This US patent is public and explains the method. As far as I know, it has never been enforced. Furthermore, this patent is associated with the paper "Going deep with convolutions", Szegedy et al. which introduced the Inception architecture, is public, very well-known, and thus I do not believe anyone would have any commercial interest in stealing it.
Furthermore, I do have the impression that the edge many companies have over their competitors is the private datasets they own much more than the architectural details.

Method and applicability
---
While the method Flush+Reload itself is not novel, its application to the DNNs case and the way to reconstruct the architecture (generating the candidates, pruning) is.
However, I do have some practical concerns about the applicability of the method.

As far as I understand, it can only work on one CPU. Most DNNs, even for inference, are run on (one or multiple) GPUs. Can the method be extended to work on GPUs?

Also, while the assumption that both the attacker and the victim use the same framework is realistic to me, I believe, they should also both use the same version of the said library, no? Otherwise some operations might be faster in some versions and slower in others, this is thus an additional and much stronger assumption to make.

At last, this would require the victim to use a public cloud service. However, as far as I know, many of the companies who could potentially design new architectures have their own private cloud. I am not certain that someone disposing of a new, private, and powerful architecture would use it on a public cloud service.

Experiments
---
The experimental section seems very limited to me. The authors show that they are able to reconstruct perfectly 2 architectures. While this is encouraging, I would like to see the limits of the proposed method.
Why not generate N random (or not so random) architectures and try to reconstruct them? Where does the method fail, where does it succeed?
What if the victim used a custom layer that the method could not recover? Does it still recover a similar architecture?


Conclusion
---
While the paper, proposed to use Flush+Reload for recovering DNNs architectures and succeeds for at least 2 non trivial architectures, I do not recommend acceptance.
First I am concerned by the problem this paper is tackling. Can this realistically happen in a real-life scenario?
Second, I am worried that the method suffers from very strong limitations in practice (eg the usage of a CPU for both victim and attacker).
Finally, and importantly, while the experiments show some interesting first results, they are limited, I am not able to judge the strengths and weaknesses of the method, and thus I cannot assess the usefulness of the proposed method.

Note: I have to say that this paper is definitely out of my area of expertise, even though I am confident in my understanding of the paper, it may be that some of my concerns are unfounded. If this is the case I will adjust my score accordingly.

**Experience Assessment:**

I do not know much about this area.

**Review Assessment: Checking Correctness Of Derivations And Theory:**

N/A

**Review Assessment: Checking Correctness Of Experiments:**

I assessed the sensibility of the experiments.

**Review Assessment: Thoroughness In Paper Reading:**

I read the paper thoroughly.

---

> ### Author Response · Authors · 2019-11-13
> **Clarification Regarding The Threat Model, Applicability, and Experiments**
>
> We thank the reviewer for the constructive feedback. Here, we provide answers to your questions and concerns, and we updated our paper for clarification.
>
> (1) Concerns about the Motivations of Our Paper
>
> Several custom architectures have outperformed the standard architectures, such as VGGs, ResNets, InceptionNets, or Transformer. For example, the Evolved Transformer [1] has achieved the state-of-the-art BLEU score of 29.8 on WMT’14 English-German translation. Also, the same architecture shows the consistent improvement over the previous architectures in four well-established language benchmarks—i.e.,  the novel architecture performs better on multiple datasets. In consequence, neural architecture search (NAS) is an active research topic in the deep learning community [6], focusing on automating the process of inventing near-optimal neural network architectures [1, 2, 3, 4, 5]. The substantial computational resources required for running NAS algorithms provide an incentive for companies and individuals to keep the networks secret to obtain a competitive advantage and therefore for attackers to steal the result of the search.
>
> (2) Applicability to GPUs
>
> Our attack is not fundamentally different for GPUs. In most deep learning frameworks, when a network performs a computation, it invokes the same function implemented in C++ and the function decides whether the back-end computation can use GPUs or not. This practice [7] maximizes the hardware compatibility of a framework; however, this also makes the framework vulnerable to our attacker who can still observe the common functions listed in Table 3 by monitoring the shared cache. On GPUs the timings would be different, so we would have to profile the computational times, e.g., the time taken for the matrix multiplication with various sizes of tensor operands. However, on both CPUs and GPUs, the computation time is proportional to the size of tensor operands, which enables our attacker to estimate the architecture parameters with timing observations. We include this information in Sec 3.1 and Appendix A of our revised paper.
>
> (3) Applicability: The Attacker and Victim Uses the Different Versions of a Framework
>
> Indeed, we implicitly assume that the attacker and victim use the same framework. We believe this is a reasonable assumption since in Machine-Learning as-a-Service scenarios (e.g., Amazon SageMaker [8] or Google’s AutoML [9]), cloud providers enforce anyone who uses these services to have the same framework version. Cloud providers want to increase the compatibility so that any network and model implemented by users (clients) can run on their environment without issues. Also, when the attacker and victim use the same framework, they are likely to use the same backend matrix multiplication library. For instance, TensorFlow uses MKL-DNN (i.e., Intel’s linear algebra library) v0.18 from Mar. 4th 2019 up to now while TensorFlow has been updated from v1.12.2 to v2.0.0. We also include this information in Sec 3.1.
>
> (4) Questions about Our Experiments
>
> [Question about Reconstructing N Random Architectures]
>
> We agree with the reviewer that generating N random architectures would be a valid experiment to perform. However, we would like to note that, even with the small search space on CIFAR10 that ENAS uses, the number of candidate networks (N) that we need to consider is 6^L times 2^(L(L−1)/2) for the network’s length (L). If we choose L=12, the total candidates for the reconstruction are over 1.6 times 10^29. Thus, even if we choose a reasonable length (L) of the victim network, reconstructing all of them would be too computationally difficult to perform.
>
> [Questions about The Attack Failure Cases]
>
> This reconstruction attack relies on several assumptions; thus, our attack successes when the assumptions are met, otherwise, the attacker fails. Here, we listed the circumstances when our assumptions break.
>
> 1) We assume that the victim architecture is implemented using the computations supported by an open-source deep learning framework. This includes that any layers a victim may create are based on these supported computations. However, if a victim uses a closed-source framework (i.e. a framework developed by and/or only available to them), it disables our attacker from monitoring computations via Flush+Reload. We believe, due to the popularity of open-source deep learning frameworks and the difficulty in developing and maintaining them, this is a rare scenario.

---

> > ### Author Response · Authors · 2019-11-13
> > **Clarification Regarding The Threat Model, Applicability, and Experiments (cont'd)**
> >
> > [Continued discussion about the reviewer's questions in the previous comment]
> >
> > 2) We also assume that the time taken for a tensor multiplication will depend on the size of tensor operands, and the computations will be performed in sequential order. If the computation time of an operation is not proportional to the size of operands, or the operations are performed out-of-order, the reconstructed architecture and its parameters will not match the victim’s architecture. However, we believe no common deep learning framework breaks these properties, and implementing them would likely be computationally inefficient.
> >
> > 3) A network with an excessive number of branches can explode the number of architecture candidates and increase the time taken for the reconstruction significantly. However, this type of architecture does not make our reconstruction attack impossible, and having this property is not common due to the significant addition to the computational overhead introduced.
> >
> > [Question about Using Custom Layers]
> >
> > Unless the custom layers are implemented by using a non-shared library, our attacker can monitor the function invocations that make up the custom layer and reconstruct the correct architecture. For instance, the authors of [10] implemented “Swish” activation in PyTorch by utilizing existing tensor operations, i.e., Swish(x) := x * Sigmoid(x). Since our attacker can monitor the multiplication (*) and Sigmoid function, the reconstruction algorithm can recover the swish activation as a sequence of a sigmoid function and a tensor multiplication.
> >
> > [References]
> > [1] So et. al, The Evolved Transformer, ICML’19
> > [2] Zoph et. al, Learning Transferable Architectures for Scalable Image Recognition, CVPR’18
> > [3] Pham et. al, Efficient Neural Architecture Search via Parameter Sharing, ICML’18
> > [4] Tan et. al, MnasNet: Platform-Aware Neural Architecture Search for Mobile, CVPR’19
> > [5] Cai et. al, ProxylessNAS: Direct Neural Architecture Search on Target Task and Hardware, ICLR’19
> > [6] 6th ICML Workshop on Automated Machine Learning, ICML’19: https://sites.google.com/view/automl2019icml/
> > [7] Customization of operations in TensorFlow: https://www.tensorflow.org/guide/create_op
> > [8] Amazon SageMaker: https://aws.amazon.com/sagemaker/
> > [9] Google AutoML: https://cloud.google.com/automl/
> > [10] Ramachandran, Prajit, Barret Zoph, and Quoc V. Le. "Swish: a Self-Gated Activation Function." arXiv preprint arXiv:1710.05941 7 (2017).

---

### Official Review · AnonReviewer2 · 2019-10-23
**Official Blind Review #2**

**Rating:** 6

**Review:**

This work proposed a method to reconstruct machine learning pipelines and network architectures using cache side-channel attack. It is based on a previous proposed method Flush+Reload that generates the raw trace of function calls. Then the authors applied several techniques to rebuild the computational graph from the raw traces. The proposed method is used to reconstruct MalConv which is a data pre-processing pipeline for malware detection and ProxyLessNas which is a network architecture obtained by NAS.

Overall, the paper is well-written and easy to read. The problem of stealing machine learning pipelines/architectures is interesting and important, since it enables an attacker to actually know the private networks that are being used for prediction. Therefore, I think this is a promising direction for future work.

I hope the authors can address my concerns as follows:

Q1: What is the knowledge of the attacker? The authors should be explicit in summarizing the detailed search space of the attacker. Currently i found it very hard to understand the capability of attacker. This is important in evaluating this work.

Q2: Can the authors add some discussion on how to defend against the proposed attack? For example, one can add some null/useless operation during execution to make the reconstruction process harder?

Q3: I am curious why the authors choose ProxylessNAS-CPU for evaluation. There is a bunch of other architectures found by NAS, e.g. MNas, ENas?

**Experience Assessment:**

I have published one or two papers in this area.

**Review Assessment: Checking Correctness Of Derivations And Theory:**

I assessed the sensibility of the derivations and theory.

**Review Assessment: Checking Correctness Of Experiments:**

I carefully checked the experiments.

**Review Assessment: Thoroughness In Paper Reading:**

I read the paper at least twice and used my best judgement in assessing the paper.

---

> ### Author Response · Authors · 2019-11-13
> **Clarifications Regarding Our Threat Model and The Potential Defenses**
>
> We thank the reviewer for the constructive feedback. Here, we provide answers to your questions and concerns, and we updated our paper for clarification.
>
> (1) About the Knowledge of Our Attacker in Reconstruction
>
> We acknowledge the reviewer’s concern that in Sec 4 where we discuss the reconstruction of MalConv and ProxylessNAS-CPU, the search space of our attacker is not clear. Here, we elaborated on the attacker’s search space, and we updated the section in our paper.
>
> [For Reconstructing Data Pre-processing Pipelines]
> Our attacker knows what tensor operations and functions to monitor in the victim’s open-source deep learning framework. These functions are model-independent; they correspond to architectural attributes designated by the deep learning framework. In Table 3, we provide a complete list of the operations and functions used for composing a data-preprocessing pipeline or a deep neural network from two popular frameworks: PyTorch and TensorFlow. We show that this knowledge is sufficient to reconstruct novel data-preprocessing pipelines, such as MalConv, that are usually shallower than the network architectures.
>
> [For Reconstructing Deep Neural Network Architectures]
> To extract the deeper network architecture automatically designed by NAS algorithms, we assume our attacker has some knowledge about the NAS search space—e.g., NASNet search space—the victim’s search process relies on. This information includes knowledge about the list of layers used and the fact that a set of layers (known as blocks) are repeatedly used—e.g., in NASNet, Normal and Reduction Blocks are used to build the architecture. We make this assumption because, from the sequence of computations observed via Flush+Reload, our attacker can easily identify a set of layers and the repetitions of the layers. However, we don’t assume how each block is composed by using the layer observations directly. Instead, we identify candidate blocks by using frequent sequence mining (FSM). We demonstrate that, under these assumptions, our attack reconstructs the ProxylessNAS architecture in 12 CPU hours rather than running a NAS algorithm from scratch that takes 40k GPU hours.
>
> We do not assume our attacker knows the victim’s architecture family. This differs from prior work [1, 2] in which the adversary already knows the victim’s network is one of the VGG family and aims to identify whether it is VGG11, 13, 16, or 19. We also do not assume our attacker is able to extract the number of for-loop iterations without noise like [1] assumed. We instead propose a technique to reduce the noise in the attacker’s observation effectively.
>
> (2) Potential Defense Mechanisms
>
> Here, we proposed four defense mechanisms that obfuscate our attacker’s observation in the Flush+Reload traces. We think this is the key idea behind what the reviewer suggested by adding null/useless operations. All the proposing defenses can apply to the deep learning framework easily; however, we also mention that the obfuscations come with a cost. We include the detailed discussion in Sec 5 of our revised paper.
>
> 1) We can add a small amount of noise (null/useless information) to the matrix multiplication to make it difficult for the attacker to estimate the computational parameters such as kernel sizes or strides. We propose to achieve it by padding zeros to tensor operands randomly. However, if our attacker can observe the same computation multiple-times, our attacker can cancel-out the blended noise and estimate the parameters correctly.
>
> 2) We can modify the victim’s architecture so that it includes the identity layers or the branches whose outputs are not used. We think a small number of null/useless operations will not increase the attacker’s computational burden significantly. This addition will increase the time needed to reconstruct the victim’s architecture by a few hours. When the defender adds an excessive amount of null/useless layers or branches, this can significantly increase the time taken for the reconstruction. However, the defense may not make the reconstruction impossible, but will increase network evaluation time significantly.
>
> 3) We can also shuffle the computation orders of a victim network; a network computes the layers sequentially as they are defined in the source code. Here, we can identify the layers that can be computed independently, and shuffle the computation orders randomly each time when the network processes an input. This will make the attacker’s observation via Flush+Reload inconsistent. However, to hold the output of independent computations, the defender may use memory more.

---

> > ### Author Response · Authors · 2019-11-13
> > **Clarifications Regarding Our Threat Model and The Potential Defenses (cont'd)**
> >
> > [Continued discussion about the potential defenses in the previous comment]
> >
> > 4) Moreover, we can run separate networks in parallel on the same physical host. The network obfuscates what our attacker will observe via Flush+Reload. In this case, our attacker may not be possible to reconstruct the victim architecture by monitoring a single query. However, if our attacker can observe multiple queries, the attacker can use the frequent sequence mining (FSM)—that we used in the block identification—to identify repeated components over the observations and can reconstruct the victim architecture.
> >
> > (3) Our Choice of ProxylessNAS-CPU Architecture
> >
> > We selected ProxylessNAS because the Proxyless (CPU) architecture provides the state-of-the-art top-1 accuracy on the ImageNet classification task, at the time of our experiments. We expect to obtain similar reconstruction results for MNAS or ENAS since they use the same tensor operations and layers in TensorFlow or PyTorch to express and run their architecture.
> >
> > [1] Yan et. al, Cache Telepathy: Leveraging Shared Resource Attacks to Learn DNN Architecture, ArXiv’18
> > [2] Duddu et. al, Stealing Neural Networks via Timing Side Channels, ArXiv’18
> > [3] Kim et. al, STEALTHMEM: System-Level Protection Against Cache-Based Side-Channel Attacks in the Cloud, USENIX’12
> > [4] Liu et. al, Catalyst: Defeating Last-level Cache Side-channel Attacks in Cloud Computing, HPCA’16

---

### Author Response · Authors · 2019-11-15
**Summary of Our Responses and Changes to the Manuscript**

We thank our reviewers again for taking the time to read, evaluate our work, and provide constructive feedback. We have uploaded a revised version of our paper, with edits to address the concerns raised. Here, we summarize our responses and updates below:

[Reviewer 1]
Q1. We provide our answer to the concerns about the motivation (in our comments).

Q2. We clarify our attack can work with GPUs (in our comments) and highlight this in Sec 3, with the detailed discussion in Appendix A (of our revised paper).

Q3. We clarify our attacker and the victim use the same framework version (in our comments) and clearly state this assumption in Sec 3.1 (of our revised paper).

Q4. We provide the answers to the questions/concerns about our experiments (in our comments).

[Reviewer 2]
Q1. We clarify the knowledge of our attacker in reconstruction (in our comments) and include this discussion in the first paragraph of Sec 4 (of our revised paper).

Q2. We discuss the potential defense mechanisms (in our comments) and include this discussion in Sec 5 (of our revised paper).

Q3. We explain the reason we choose ProxylessNAS-CPU for evaluation (in our comments)

[Reviewer 3]
Q1. We provide our attacker’s gain in terms of time and resources compared to running NAS from scratch (in our comments) and include this information in Sec 4 (of our revised paper).

Q2/3. We provide a discussion about the limitations of our attack (in our comments).

Please see our replies to each reviewer for our detailed responses to individual points.

---

### Decision · Program_Chairs · 2019-12-19

**Decision:**

Accept (Poster)

**Comment:**

This paper proposes using Flush+Reload to infer the deep network architecture of another program, when the two programs are running on the same machine (as in cloud computing or similar).

There is some disagreement about this paper; the approach is thoughtful and well executed, but one reviewer had concerns about its applicability and realism. Upon reading the author's rebuttal I believe these to be largely addressed, or at least as realistically as one can in a single paper. Therefore I recommend acceptance.